# Clinical Predictors for Abnormal ALT in Patients Infected with COVID-19—A Retrospective Single Centre Study

**DOI:** 10.3390/pathogens12030473

**Published:** 2023-03-16

**Authors:** Wei Da Chew, Jonathan Kuang, Huiyu Lin, Li Wei Ang, Wei Lyn Yang, David C. Lye, Barnaby E. Young

**Affiliations:** 1Department of Gastroenterology & Hepatology, Tan Tock Seng Hospital, 11 Jalan Tan Tock Seng, Singapore 308433, Singapore; 2National Centre for Infectious Diseases, 16 Jalan Tan Tock Seng, Singapore 308442, Singapore; 3Lee Kong Chian School of Medicine, Nanyang Technological University, Singapore 636921, Singapore

**Keywords:** COVID-19, ALT, R-factor, hypoxia, liver test

## Abstract

Objective: Abnormal liver tests have been associated with worse clinical outcomes in patients infected with COVID-19. This retrospective observational study from Singapore aims to elucidate simple clinical predictors of abnormal alanine aminotransferase (ALT) in COVID-19 infections. Design: 717 patients hospitalised with COVID-19 at the National Centre for Infectious Diseases (NCID), Singapore, from 23 January–15 April 2020 were screened, of which 163 patients with baseline normal alanine transferase (ALT) and at least two subsequent ALTs performed were included in the final analysis. Information on baseline demographics, clinical characteristics and biochemical laboratory tests were collected. Results: 30.7% of patients developed abnormal ALT. They were more likely to be older (60 vs. 55, *p* = 0.022) and have comorbidities of hyperlipidaemia and hypertension. The multivariate logistic regression showed that R-factor ≥1 on admission (adjusted odds ratio (aOR) 3.13, 95% Confidence Interval (CI) 1.41–6.95) and hypoxia (aOR 3.54, 95% CI 1.29–9.69) were independent risk factors for developing abnormal ALT. The patients who developed abnormal ALT also ran a more severe course of illness with a greater proportion needing supplementary oxygen (58% vs. 18.6%, *p* < 0.0005), admission to the Intensive Care Unit (ICU)/High Dependency Unit (HDU) (32% vs. 11.5%, *p* = 0.003) and intubation (20% vs. 2.7%, *p* < 0.0005). There was no difference in death rate between the two groups. Conclusions: Liver injury is associated with poor clinical outcomes in patients with COVID-19. R-factor ≥1 on admission and hypoxia are independent simple clinical predictors for developing abnormal ALT in COVID-19.

## 1. Introduction

Coronavirus disease-19 (COVID-19) is an ongoing pandemic posing a health threat especially for unvaccinated individuals. There are a total of 660 million infections and 6 million deaths across nearly 200 countries as of January 2023 [1]. The world is still grappling with waves of infection caused by contagious COVID-19 variants despite the progress made by vaccination programs and public health measures. Most infected persons remain asymptomatic or have mild respiratory symptoms, but around 15% will develop severe pulmonary disease typically over 10 days, which might progress to multi-organ failure and even death [2]. Concomitant hepatic involvement is well-recognised and liver abnormalities have been reported in up to 4–33% in Chinese studies and up to 58% in the largest US series of patients with COVID-19 [2,3,4,5]. Unvaccinated individuals have an increased risk of developing severe COVID-19 infection with up to 30% of the world still unvaccinated [6]. Furthermore, unvaccinated individuals account for up to 70% of the population in low-income countries, thus, simple clinical predictors will be helpful in addressing the impact of this vaccine inequality [7].

Studies have reported that raised ALT was associated with poorer clinical outcomes including severity of the disease, longer hospital stays, admission to ICU and mortality [4,8,9,10,11,12]. However, there are limited studies that have looked at factors predisposing to the de novo development of ALT abnormalities during the clinical course of the disease, which has a significant impact on the prognostic outcome [13]. Thus, our study aimed to identify clinical predictors associated with the de novo development of ALT abnormalities after COVID-19 infection.

## 2. Methods

We performed a retrospective observational study of 717 adult patients with COVID-19 admitted to NCID (Singapore) between 23 January and 15 April 2020. All patients were confirmed to have COVID-19 by a positive SARS-CoV-2 real-time reverse transcriptase–polymerase chain reaction as per the World Health Organization (WHO) interim guidance [14]. The QIAamp viral RNA mini kit (Qiagen Hilden, Germany) was used for sample RNA extraction. At the time of the study, all patients with confirmed COVID-19 were admitted to hospital for isolation regardless of disease severity. Waiver of informed consent for retrospective collection of clinical data from infected individuals by review of medical records was granted by the local ethics committee (Domain Specific Review Board, ref 2020/01122).

### 2.1. Study Design

The patients’ demographics, medical history, laboratory tests, radiological reports and clinical outcomes were obtained from electronic medical records. Data were collected until 30 April 2020 using the Research Electronic Data Capture Software. Symptoms including onset of infection, diarrhoea, abdominal pain and vomiting were extracted. Medical history included hypertension, hyperlipidemia, diabetes mellitus, ischemic heart disease and chronic liver disease (CLD). The following events during hospitalisation were collected: the requirement of any supplemental oxygen, mechanical ventilation, High Dependency Unit (HDU)/ICU admission and death. For analysis on the risk factors contributing to the development of abnormal ALT (Figure 1), only patients with a normal ALT at baseline and with at least one subsequent measured ALT were included. Baseline clinical characteristics of this subgroup were reported in Table 1 and details regarding the type of drugs administered during hospitalization and their initiation date were collected.

### 2.2. Definitions

COVID-19-associated liver injury is defined as any raised ALT in patients with COVID-19 with or without pre-existing liver disease [15]. The upper limit of normal (ULN) ALT was defined according to the criteria of The Asian-Pacific Association for the Study of the Liver (40 U/L for both genders) [16]. The ULN of other laboratory tests was defined as per the local laboratory reference ranges. De novo liver test abnormalities were defined as the development of abnormal ALT in patients with normal ALT at admission. The R-factor was calculated as (ALT/ULN of ALT)/(ALP/ULN of alkaline phosphatase (ALP)) [16]. Hypoxia was defined as requiring any supplemental oxygen when room air saturation dropped below 92%, as measured by pulse oximeter. The decision to initiate and/or stop medications was at the treating physicians’ discretion following NCID treatment protocol.

### 2.3. Statistical Analysis

Numbers and proportions were presented for categorical variables, and the median and interquartile range (IQR) for continuous variables. Fisher’s exact test or the chi-square test was used to compare categorical variables and the Mann–Whitney U test to compare continuous variables between patients who developed abnormal ALT and those who remained normal.

An “optimal” cutpoint for the binary coding of R factor on admission was estimated by testing which cutpoint yields the best discrimination based on Youden’s index (sensitivity + specificity −1); this index was used as the metric to maximize for predicting the outcome of the development of abnormal ALT. The “optimal” cutpoint was 0.933, which resulted in a Youden’s index of 0.3041. However, the estimation of optimal cutpoints in a specific sample may lead to highly variable “optimal” cutpoints and a systematic over-estimation of the out-of-sample performance. To address these concerns, we used a robust approach by drawing 1000 bootstrap samples of the same size as our study sample (163 patients) and conducting cutpoint optimization for maximizing Youden’s index on every bootstrapped sample. The mean of the “optimal” cutpoints on the bootstrap samples was 1.0039 which yielded a Youden’s index of 0.286 on our study sample (sensitivity was 64% and specificity was 65%). Hence, we used the cutpoint of 1.0 for the categorization of R factor on admission.

We used multivariable logistic regression to elucidate risk factors associated with the development of abnormal ALT among COVID-19 patients with normal ALT at baseline. Covariates with *p* < 0.10 in the univariable logistic regression analyses were included in the multivariable model. All statistical tests were two-sided and statistical significance was taken as *p* < 0.05. Statistical analyses were performed using IBM SPSS Statistics for Windows, version 24 (IBM Corp, Armonk, NY, USA) and R version 3.6.2 (R foundation for Statistical Computing, Vienna, Austria).

## 3. Results

Data were available for 717 patients with COVID-19 admitted to NCID, and 48 subjects with no liver function tests (LFT) were excluded (Figure 1). Among the 669 patients with COVID-19, 518 (77.4%) had normal baseline ALT. Of these 518 patients with a normal baseline ALT, a sub-group of 163 (31.5%) patients, who had ≥2 ALT tests carried out, were included in the final analysis.

The baseline demographics and clinical characteristics of the 163 patients with ≥2 ALT tests, stratified by normal and abnormal ALT, are shown in Table 1.

The median age was 56 years old, 59% were male and 60% were Chinese. Patients who went on to develop abnormal ALT were more likely to be older and have comorbidities of hyperlipidaemia and hypertension. Of the analysis of 163 patients, a higher proportion of patients who developed abnormal ALT required beta-lactam antibiotics (44.0% vs. 22.1%, *p* = 0.08), lopinavir–ritonavir (32.0% vs. 8.0%, *p* < 0.0005) and interferon beta 1b (12.0% vs. 2.7%, *p* = 0.025). Patients who went on to develop abnormal ALT had a higher ALT level on admission (29 vs. 21, *p* < 0.0005), higher ALT/LDH ratio (0.06 vs. 0.05; *p* = 0.039), R-factor (1.15 vs. 0.87; *p* < 0.0005) and C-reactive protein (CRP) (30.1 vs. 6.85, *p* < 0.0005).

The R-factor on admission was noted to trend higher for the development of abnormal liver tests regardless of the day of illness when compared to those whose liver tests remained normal. (Figure 2)

The patients who developed abnormal ALT also ran a more severe course of illness with a greater proportion needing supplementary oxygen (58% vs. 18.6%, *p* < 0.0005), admission to the Intensive Care Unit (ICU)/High Dependency Unit (HDU) (32% vs. 11.5%, *p* = 0.003) and intubation (20% vs. 2.7%, *p* < 0.0005). There was no difference in death rate between the two groups.

On univariable logistic regression analyses (Table 2), the use of beta-lactam antibiotics, lopinavir–ritonavir, interferon-beta1b, the presence of hypoxia and R factor on admission were associated with the development of abnormal ALT. In the multivariable logistic regression, only patients with R-factor ≥1 on admission (aOR 3.13, 95% CI 1.41–6.95) and hypoxia (aOR 3.54, 95% CI 1.29–9.69) were at a higher odds of developing abnormal ALT.

## 4. Discussion

Our study seeks to elucidate simple clinical predictors in the development of abnormal ALT. In contrast to other studies which only reviewed blood results on admission, we obtained serial data for liver tests that were temporally correlated with clinical outcomes and risk factors. About 31.3% of our patients developed liver test abnormalities, which were predominantly raised ALT. This is similar to other studies where 25–30% of patients with COVID-19 in Asia had abnormal liver tests [17], whereas the proportion reached as high as 39–58% in the Western population [5,10,18]. ALT elevation in our group of patients was mostly mild and self-limiting, consistent with many other studies [11,15,19]. Previously, the significance and prognostic value of LFT abnormalities were uncertain, but it is increasingly accepted that liver test abnormalities correlate to disease severity and poor outcomes of COVID-19 [17,20]. A meta-analysis of 16 retrospective cohort studies based on Chinese and Western studies [3,10,21] showed that elevated levels of liver injury markers, particularly aminotransferases, may be associated with progression to severe disease or death. Death in our cohort was not significant, probably due to our very low fatality rate of 0.05% (28 deaths), as of 1 November 2020.

Transaminase elevation during COVID-19 is mostly mild and reversible and elevations are usually <5 times the upper limit of normal in up to 80% of infected patients; however, severe ALI (acute liver injury) and/or acute liver failure, though rare, has been reported [12]. As the clinical manifestations of early liver dysfunction are not apparent, it may thus be worthwhile to identify simple risk factors for the development of abnormal liver tests, which in turn have a prognostic value. Wang et al. looked at 657 patients with COVID-19 and found that more patients with liver injury than without had increased serum IL-2R, TNFα, ferritin, higher serum high-sensitivity C-reactive protein (hsCRP), PCT, ESR, γ-GT and LDH [22]. Multivariate regression analysis revealed that the increasing odds of liver injury were related to male, hsCRP (≥10 mg/L) and the neutrophil-to-lymphocyte ratio (NLR) (≥5). Other papers have also found some risk factors—for example, plateletcrit, retinol-binding protein and carbon dioxide combining power could predict liver function damage in patients with a moderate COVID-19 infection [23]. However, the above-mentioned parameters are not widely available and are cost-ineffective in resource-limited low-income countries.

We found that R-factor on admission may be used to predict the development of abnormal ALT. R-factor was initially developed to assess liver injury in drug-induced liver injury (DILI) but is frequently used in clinical practice to objectively define different patterns of liver injury [8]. Although the admission ALT values were within the normal laboratory reference range, there were subtle differences in their presenting levels: relatively higher ALT levels on admission may have a predictive value for developing abnormal ALT. In our analysis of 163 patients with normal baseline ALT, those with an R-factor of ≥1 on admission had a three-fold risk of developing ALT abnormalities. Thus, R-factor—which is a simple-to-calculate parameter—can be consider for use on admission to predict the development of abnormal ALT.

The specific mechanism involved in acute hepatic injury in patients with COVID-19 remains unclear, but it is likely multifaceted and multifactorial [12,24].SARS-CoV-2 gains access into the host through the angiotensin-converting enzyme 2 (ACE2) receptor which is present in abundance in cholangiocytes and lesser in hepatocytes. Sun et al. [15] described several possible mechanisms of liver damage in COVID-19 including hypoxic hepatitis and the direct cytotoxicity of SARS-CoV-2 virus in hepatocytes. SARS-CoV-2 infection has also been associated with a highly unusual and overt proinflammatory ‘cytokine storm’, suggesting that this tremendous proinflammatory attack contributes to observed early and late hepatologic abnormalities [24]. In a large cohort of patients with COVID-19, systemic inflammation—as detected by increased levels of IL-6, hsCRP or ferritin—was highly correlated with the degree of ALI, as assessed by AST levels [25,26]. Others have reported non-specific histological findings on liver biopsies in patients with COVID-19, demonstrating microvesicular steatosis, sinusoidal dilatation and occasional necrosis [27,28].

Drugs used in the treatment of COVID-19 have been described as a possible cause of liver injury, with antivirals and corticosteroids being associated with ALT/AST elevation [29]. ALT and bilirubin levels were more likely to rise after starting antiviral treatment, while dropping gradually after the discontinuation of antivirals, with stable ALP levels throughout [11,29]. In contrast, our study showed that medications including antivirals/acetaminophen were not independently associated with the development of abnormal ALT on multivariable analysis.

The lung is the most significant site of involvement for COVID-19, with 20–40% of patients with COVID-19 having varying degrees of hypoxemia [4,9,13,20]. We found that hypoxia was one of the risk factors which was independently associated with the development of ALT abnormality. The hypoxia of respiratory distress syndrome can result in the continuous generation of reactive oxygen species and various pro-inflammatory factors to induce liver damage [28]. Other potential mechanisms of liver injury in hypoxia include hepatic venous congestion secondary to elevated central venous pressure, hypoxia reperfusion injury, high positive end-respiratory pressure (PEEP) use in intubated patients and vascular alterations [3,30,31].

This is the first study from Singapore looking at LFT abnormalities and its clinical impact in patients with COVID-19 (unvaccinated cohort). Our study is one of the few to use the R-factor derived from liver tests as an objective, has an early and simple way to risk-stratify patients who will develop abnormal ALT and may have a role in prognosticating and predicting clinical severity in combination with other established predictors, such as LDH and CRP [32]. This is especially important in countries with limited healthcare resources coupled with a low vaccination rate.

Our research has some limitations. Firstly, it was retrospective, and most cases did not have liver tests prior to COVID-19 infection. Data were incomplete regarding prior liver disease such as viral hepatitis, frequency of non-alcoholic fatty liver disease, concomitant supplement use and alcohol history based on the retrospective analysis of available medical records, and thus, are not included in formal analysis. Our institution’s standard liver test panel did not include AST/gamma-glutamyl transferase (GGT) and liver tests were also performed at variable intervals based on the clinician’s judgement. Thus, selection bias may exist as sequential liver tests might be ordered more frequently in unwell patients. Finally, there were no liver biopsies conducted, as such, we were unable to prove with histology whether cytopathic effects of COVID-19 on the liver were indeed present [32].

## 5. Conclusions

The R-factor ratio has a role in the risk stratification of COVID-19 infection as a clinical predictor, as higher values were associated with an increased risk of the de novo development of ALT abnormalities. While liver injury has been postulated to be multifactorial including direct cytotoxicity and DILI, our study found that patients with both hypoxia and an R-factor of ≥1 had increased odds of developing abnormal ALT. The pathogenic effects are possibly driven by immune and inflammatory responses rather than direct cytotoxicity or DILI.

## Figures and Tables

**Figure 1 pathogens-12-00473-f001:**
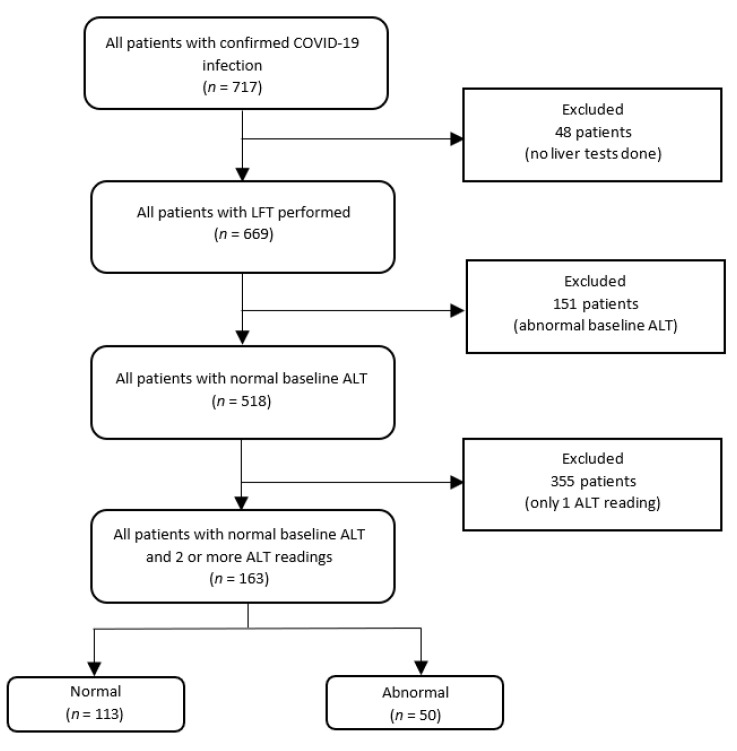
Selection of patients with normal baseline ALT with subsequent 2 or more ALT reading.

**Figure 2 pathogens-12-00473-f002:**
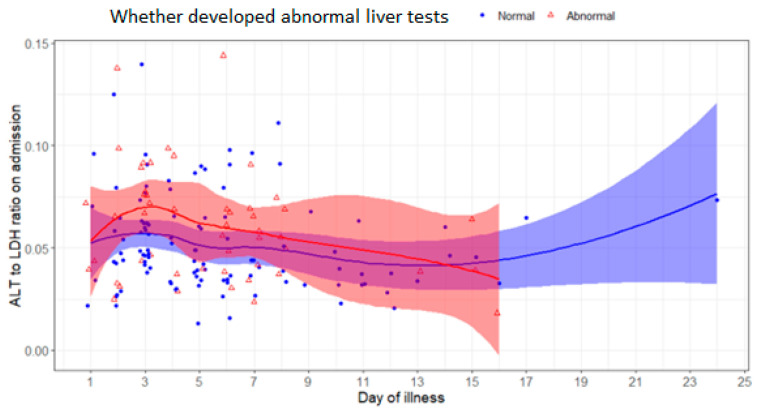
Scatterplot of R factor on admission and day of illness among 163 patients with COVID-19 according to whether patient developed abnormal liver tests or not.

**Table 1 pathogens-12-00473-t001:** Demographics, comorbidities, laboratory investigations and clinical outcomes of patients with COVID-19 stratified by ALT.

Characteristics	All (*n* = 163)	Status of ALT		*p*-Value
		Abnormal (*n* = 50)	Normal (*n* = 113)	
Age in years, median (IQR)	56 (43–65)	60 (50–67)	55 (37–64)	**0.022**
Gender, *n* (%)				0.124
Male	96 (58.9)	34 (68.0)	62 (54.9)	
Female	67 (41.1)	16 (32.0)	51 (45.1)	
Ethnic group, *n* (%)				0.520
Chinese	98 (60.1)	34 (68.0)	64 (56.6)	
Malay	18 (11.0)	4 (8.0)	14 (12.4)	
Indian	20 (12.3)	6 (12.0)	14 (12.4)	
Others	27 (16.6)	6 (12.0)	21 (18.6)	
Comorbidities, *n* (%)				
Diabetes	32 (19.6)	13 (26.0)	19 (16.8)	0.201
Hyperlipidemia	57 (35.0)	24 (48.0)	33 (29.2)	**0.032**
Hypertension	61 (37.4)	26 (52.0)	35 (31.0)	**0.014**
Ischemic heart disease	15 (9.2)	7 (14.0)	8 (7.1)	0.238
Chronic liver disease	4 (2.5)	1 (2.0)	3 (2.7)	1.000
Charlson Comorbidity Index, median (IQR)	0 (0–1)	0 (0–1)	0 (0–1)	0.400
BMI, kg/m^2^, median (IQR), *n* = 46	24.3 (23.2–27.9)	22.9 (22.1–24.2)	24.6 (23.6–28.7)	**0.011**
GI symptoms, *n* (%)				
Diarrhoea	29 (17.8)	12 (24.0)	17 (15.0)	0.186
Abdominal pain	4 (2.5)	0 (0.0)	4 (3.5)	0.313
Nausea/vomiting	10 (6.1)	0 (0.0)	10 (8.8)	**0.032**
Abnormal chest radiography on admission	55 (33.7)	22 (44.0)	33 (29.2)	0.074
Laboratory investigations on admission, median (IQR)				
ALT, U/L	23 (18–31)	29 (22–33)	21 (17–26)	**<0.0005**
ALT/LDH ratio, *n* = 162	0.05 (0.04–0.07)	0.06 (0.04–0.07)	0.05 (0.03–0.06)	**0.039**
ALP	72 (60–89)	72 (61–90)	72 (60–89)	0.700
R factor	0.94 (0.70–1.26)	1.15 (0.86–1.49)	0.87 (0.63–1.19)	**<0.0005**
WBC, ×10^9^/L	4.70 (3.80–5.70)	4.75 (3.80–5.83)	4.70 (3.85–5.70)	0.844
Lymphocyte, ×10^9^/L	1.11 (0.84–1.49)	0.99 (0.74–1.23)	1.20 (0.87–1.65)	**0.002**
PLT, ×10^9^/L	188 (150–225)	177 (142–223)	193 (155–226)	0.306
CRP, mg/L, *n* = 162	10.75 (3.15–39.40)	30.10 (11.28–50.65)	6.85 (1.95–23.88)	**<0.0005**
LDH, U/L, *n* = 162	420 (350–547)	482 (378–572)	408 (342–525)	**0.033**
Creatinine, μmol/L	72 (61–87)	76 (65–88)	71 (59–87)	0.288
Albumin, g/L, *n* = 156	39 (37–42)	39 (37–41)	40 (37–43)	**0.044**
BIL, μmol/L, *n* = 152	11 (9–14)	11 (9–14)	12 (9–15)	0.555
Use of paracetamol, *n* (%)				0.125
No	15 (9.2)	3 (6.0)	12 (10.6)	
Yes, <2 g/day	100 (61.3)	27 (54.0)	73 (64.6)	
Yes, ≥2 g/day	48 (29.4)	20 (40.0)	28 (24.8)	
Medication used, *n* (%)				
NSAIDs	22 (13.5)	4 (8.0)	18 (15.9)	0.218
β-lactam	47 (28.8)	22 (44.0)	25 (22.1)	**0.008**
Hydroxychloroquine	7 (4.3)	1 (2.0)	6 (5.3)	0.677
Lopinavir/Ritonavir (Kaletra)	25 (15.3)	16 (32.0)	9 (8.0)	**<0.0005**
Remdesivir	12 (7.4)	5 (10.0)	7 (6.2)	0.516
Interferon	9 (5.5)	6 (12.0)	3 (2.7)	**0.025**
Days of symptoms before admission, median (IQR)	4 (3–7)	4 (2–7)	5 (3–7)	0.396
Length of stay in days, median (range)	13 (8–17)	16 (13–24)	11 (7–16)	**<0.0005**
Clinical severity				
HDU/ICU, *n* (%)	29 (17.8)	16 (32.0)	13 (11.5)	**0.003**
Required supplementary oxygen, *n* (%)	50 (30.7)	29 (58.0)	21 (18.6)	**<0.0005**
Days on supplementary oxygen, median (IQR), *n* = 50	11 (6–18)	12 (6–21)	8 (5–15)	0.151
Intubated, *n* (%)	13 (8.0)	10 (20.0)	3 (2.7)	**<0.0005**
Death, *n* (%)	5 (3.1)	3 (6.0)	2 (1.8)	0.169

Sample size, *n* = 163, except where indicated. *p* values are from Fisher’s exact test or chi-square test for categorical variables and the Mann–Whitney U test for continuous variables. *p* values <0.05 are in bold. ALP, alkaline phosphatase; ALT, alanine aminotransferase; AST, aspartate aminotransferase; BIL, bilirubin; BMI, body mass index; CRP, c-reactive protein; GI, gastrointestinal; ICU, intensive care unit; IQR, interquartile range; LDH, lactate dehydrogenase; HDU, high dependency unit; PLT, platelet count; WBC, white blood cell.

**Table 2 pathogens-12-00473-t002:** Odds ratios of risk factors for the development of abnormal ALT among COVID-19 patients with normal ALT at baseline.

Variable	Univariable Model			Multivariable Model ^‡^		
	cOR	(95% CI)	*p* Value	aOR	(95% CI)	*p* Value
Age in years						
<45	1.00	Referent		1.00	Referent	
45–64	3.42	(1.28–9.11)	0.014	2.67	(0.84–8.47)	0.096
65+	4.31	(1.49–12.42)	0.007	2.84	(0.66–12.19)	0.160
Gender						
Male	1.00	Referent				
Female	0.57	(0.28–1.15)	0.118			
Diabetes	1.74	(0.78–3.87)	0.176			
Hyperlipidemia	2.24	(1.13–4.45)	0.022	1.14	(0.43–3.00)	0.796
Hypertension	2.41	(1.22–4.78)	0.011	0.89	(0.31–2.58)	0.835
Ischemic heart disease	2.14	(0.73–6.26)	0.166			
Presence of GI symptom(s) on admission	1.17	(0.53–2.58)	0.695			
Abnormal chest X-ray on admission	1.90	(0.96–3.80)	0.067	0.91	(0.36–2.25)	0.833
R factor on admission						
<1	1.00	Referent		1.00	Referent	
≥1	3.12	(1.56–6.24)	**0.001**	3.13	(1.41–6.95)	**0.005**
Use of acetaminophen						
No	1.00	Referent				
Yes, <2 g/day	1.48	(0.39–5.65)	0.567			
Yes, ≥2 g/day	2.86	(0.71–11.46)	0.139			
Use of β-lactam	2.77	(1.35–5.65)	**0.005**	1.12	(0.38–3.24)	0.840
Use of Hydroxychloroquine	0.36	(0.04–3.11)	0.355			
Use of Lopinavir/Ritonavir (Kaletra)	5.44	(2.20–13.43)	**<0.0005**	2.20	(0.57–8.45)	0.252
Use of Remdesivir	1.68	(0.51–5.58)	0.395			
Use of interferon	5.00	(1.20–20.88)	**0.027**	0.80	(0.12–5.22)	0.813
Hypoxia	6.05	(2.90–12.62)	**<0.0005**	3.54	(1.29–9.69)	**0.014**

^‡^ Variables in the multivariable logistic regression model were age group, hyperlipidemia, hypertension, whether there was abnormal chest X-ray on admission, R factor on admission, use of β-lactam, use of LPV/r, use of interferon and hypoxia. *p* values < 0.05 are in bold. aOR, adjusted odds ratio, cOR, crude odds ratio.

## Data Availability

The data presented in this study are available on request from the corresponding author.

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
