# Peer review of "Clinical Predictors for Abnormal ALT in Patients Infected with COVID-19—A Retrospective Single Centre Study"

_pathogens, 2023, doi:10.3390/pathogens12030473_

Round 1

Reviewer 1 Report

Your observaiton in patients with COVID-19, focusing on the significance of ALT to anticipate the outcome of the disease is so interesting and important.

The increase of ALT level in patients with COVID-19 seems good indicator of outcome, possible risk of complications. As one of clinical experience, to report this manuscript seems valuable for readers.Indeed, previous reports had already reported that the liver function test seemed good indicator to be treated in ICU. This report indicated many valuable factors including oxygen administration requirement, ICR treatment and also hyperlipidemia and hypertension but not diabetes nor ischemic heart disease. No further improvements neede, as long as they could take number of patients, the outcome and analysis should be appropriate as it was.
The references are appropriate. This clinical report is indeed one of clinical reports involving COVID-19..

Author Response

Thank you for kind comments. No changes were requested by the reviewer. 

Reviewer 2 Report

This is a retrospective observational study from Singapore aimed to assess the relationship between abnormalities of liver tests and clinical outcomes in patients with Covid-19. The investigators screened 669 patients with Covid-19, 518 had normal ALT levels at baseline. 163 (30.7%) of 518 had two or more subsequent ALT readings. The authors observed that 50 (30.7%) of 163 patients had abnormal ALT values.  

Patients with abnormal ALT status had greater age (P=0.02) and higher frequency of GI symptoms (P=0.032). A greater rate of comorbidities (arterial hypertension, hyperlipidemia) was found in patients with abnormal ALT levels. According to univariate analysis, the R factor and clinical severity at admission (intubation rate, ICU/HDU status, supplementary oxygen) were greater in the subset of patients with raised abnormal ALT levels.  

Multivariate analysis (logistic regression analysis) reported that hypoxia and R-factor had independent and significant relationship with abnormal ALT status. Of note, there was no significant difference with regard to death rate between COVID-19 patients who developed raised ALT values and those who had persistent normal ALT.

An important shortcoming of the study is that there are no data on viral hepatitis, frequency of NAFLD (non alcoholic fatty liver disease), alcohol abuse or concomitant assumption of agents responsible of liver damage. 

The authors in the Section Introduction affirm that ALT values are liver function tests. This is incorrect.

Liver function tests include prothrombin time (PT/INR), activated partial thromboplastin time (aPTT), and albumin. AST/ALT are markers of liver injury (cellular integrity), and other tests are associated with abnormalities of the biliary tract (alkaline phosphatase and gamma-glutamyl transpeptidase).

It is better to call these liver chemistries or liver tests rather that liver  function tests. 

Author Response

Thank you for the detailed review of our paper and the constructive comments 

In response to the feedback: 

  1. An important shortcoming of the study is that there are no data on viral hepatitis, frequency of NAFLD (non alcoholic fatty liver disease), alcohol abuse or concomitant assumption of agents responsible of liver damage 
    - We have edited and included this in our discussions/limitations

2.  Authors in the Section Introduction affirm that ALT values are liver function tests. This is incorrect.
....
It is better to call these liver chemistries or liver tests rather that liver function tests. 

  • we have changed relevant terminology to "liver tests" in the paper

Thank you once again for your comments